# Strain-engineering Mott-insulating $La_2CuO_4$

O. Ivashko [1], M. Horio [1], W. Wan[2], N.B. Christensen[2], D.E. McNally[3], E. Paris[3], Y. Tseng[3], N.E. Shaik[4], H.M. Rønnow [4], H.I. Wei [5], C. Adamo[6], C. Lichtensteiger[7], M. Gibert[1], M.R. Beasley[6], K.M. Shen[5], J.M. Tomczak [8], T. Schmitt[3] & J. Chang [1]

The transition temperature $T_c$ of unconventional superconductivity is often tunable. For a monolayer of FeSe, for example, the sweet spot is uniquely bound to titanium-oxide substrates. By contrast for $La_{2-x}Sr_xCuO_4$ thin films, such substrates are sub-optimal and the highest $T_c$ is instead obtained using $LaSrAlO_4$. An outstanding challenge is thus to understand the optimal conditions for superconductivity in thin films: which microscopic parameters drive the change in $T_c$ and how can we tune them? Here we demonstrate, by a combination of x-ray absorption and resonant inelastic x-ray scattering spectroscopy, how the Coulomb and magnetic-exchange interaction of $La_2CuO_4$ thin films can be enhanced by compressive strain. Our experiments and theoretical calculations establish that the substrate producing the largest $T_c$ under doping also generates the largest nearest neighbour hopping integral, Coulomb and magnetic-exchange interaction. We hence suggest optimising the parent Mott state as a strategy for enhancing the superconducting transition temperature in cuprates.

[1] Physik-Institut, Universität Zürich, Winterthurerstrasse 190, CH-8057 Zürich, Switzerland. [2] Department of Physics, Technical University of Denmark, DK-2800 Kongens Lyngby, Denmark. [3] Photon Science Division, Swiss Light Source, Paul Scherrer Institut, CH-5232 Villigen PSI, Switzerland. [4] Institute of Physics, École Polytechnique Fedérale de Lausanne (EPFL), CH-1015 Lausanne, Switzerland. [5] Department of Physics, Laboratory of Atomic and Solid State Physics, Cornell University, Ithaca, NY 14853, USA. [6] Department of Applied Physics, Stanford University, Stanford, CA 94305, USA. [7] Department of Quantum Matter Physics, University of Geneva, 24 Quai Ernest Ansermet, 1211 Geneva, Switzerland. [8] Institute of Solid State Physics, Vienna University of Technology, A-1040 Vienna, Austria. These authors contributed equally: O. Ivashko, M. Horio. Correspondence and requests for materials should be addressed to O.I. (email: oleh.ivashko@physik.uzh.ch) or to J.C. (email: johan.chang@physik.uzh.ch)

Exposed to pressure, the lattice parameters of a material generally shrink. In turn, the electronic nearest neighbour hopping integral $t$ increases, due to larger orbital overlap. In a Mott insulator this enhancement can trigger a bandwidth-controlled insulator-to-metal transition[1]. Indeed, the ratio, $U/t$, of the electron-electron (Coulomb) interaction $U$ and the hopping $t$ may be driven below its critical value. This premise has led to prediction of a pressure-induced insulator-to-metal transition in hypothetical solid hydrogen[2]. Experimentally, pressure-induced metallisations have been realised, e.g., in $NiS_2$[3] and organic salts[4]. However, besides its impact on the bandwidth, pressure also influences (in a complex fashion) the electron-electron interaction $U$—an effect that has received little attention so far. The fate of Mott insulators exposed to external pressure therefore remains an interesting (and unresolved) problem to consider.

In the case of layered copper-oxide materials (cuprates), superconductivity emerges once the Mott insulating state is doped away from half-filling[5]. In fact, it is commonly believed that the Mott state is a precondition for cuprate high-temperature superconductivity. While the optimal doping has been established for all known cuprate systems, the ideal configuration—for superconductivity—of the parent Mott state has not been identified. Typically, it is reported that hydrostatic pressure has a positive effect on $T_c$[6,7]. However, the microscopic origin of this finding remains elusive. In particular, how pressure influences the local Coulomb interaction $U$ and the inter-site magnetic-exchange interaction—to lowest order—$J_{eff} = 4t^2/U$, is an unresolved problem.

Here we present a combined x-ray absorption spectroscopy (XAS) and resonant inelastic x-ray scattering (RIXS) study of the $La_2CuO_4$ Mott insulating phase. We show that by straining thin films, the crystal field environment, as well as the energy scales $t$ and $U$ that define the degree of electronic correlations, can be tuned. In stark contrast to predictions for elementary hydrogen[2] and observations on standard Mott insulating compounds[3,4], we demonstrate that $U/t$ remains approximately constant with in-plane strain. In $La_2CuO_4$, both $U$ and $t$ are increasing with compressive strain. In-plane strain is therefore not pushing $La_2CuO_4$ closer to the metallisation limit. Instead, strain enhances the stiffness, i.e., the exchange interaction $J_{eff}$, of the antiferromagnetic ordering. These experimental observations are consistent with our band structure and constrained Random Phase Approximation (cRPA) calculations that reveal the same trends for $t$, $U$ and $J_{eff}$. For superconductivity, originating from the anti-ferromagnetic pairing channel, the exchange interaction is a key energy scale. Our study demonstrates how $J_{eff}$ can be controlled and enhanced with direct implications for the optimisation of superconductivity.

## Results

**Crystal-field environment.** Thin films (8–19 nm) of $La_2CuO_4$ grown on substrates with different lattice parameters are studied. In this fashion both compressive [$LaSrAlO_4$ (LSAO)] and tensile [$NdGaO_3$ (NGO), $(LaAlO_3)_{0.3}(Sr_2TaAlO_6)_{0.7}$ (LSAT) and $SrTiO_3$ (STO)] strain is imposed. Strain is defined by $\varepsilon \equiv (a - a_0)/a_0$ where $a$ and $a_0$ are the in-plane lattice parameters of the thin film and the bulk, respectively. For the above-mentioned samples, $\varepsilon = -1.25$, 1.59, 1.70 and 2.67% is obtained, respectively. The substrates are tuning both the in-plane and out-of-plane lattice parameters of the $La_2CuO_4$ films (see Table 1), directly affecting the electronic energy scales of the system. This can be readily observed from the XAS spectra at the copper $L_3$ edge [see Fig. 1a]. A considerable shift (~280 meV) of the Cu $L_3$ edge is found when comparing the compressive strained LCO/LSAO with the tensile strained LCO/STO film. The $dd$ excitations probed through the Cu $L_3$ edge, exhibit a similar systematic shift [see Fig. 1b–d]. The strain-dependent line shape of the $dd$ excitations, points to a change in the crystal-field environment. The double peak structure, known for bulk $La_2CuO_4$[8,9], is also found in our thin films on LSAO, NGO and LSAT substrates. For LCO/STO, however, a more featureless line shape is found, resembling doped $La_{2-x}Sr_xCuO_4$ (LSCO)[10]. All together, the shift of the Cu $L_3$ edge and the centre of mass [Fig. 1d] of the $dd$ excitations (along with the line-shape evolution) demonstrate the effectiveness of epitaxial strain for tuning the electronic excitations.

**Zone-boundary magnons.** The low-energy part of the RIXS spectra in the vicinity to the high-symmetry zone-boundary points (1/2, 0) and (1/4, 1/4) are shown in Fig. 1e,f. Generally, the spectra are composed of elastic scattering, a magnon and a weaker multimagnon contribution on a weak smoothly-varying background. In all zone-boundary (ZB) spectra, the magnon excitation is by far the most intense feature. The ZB magnon excitation energy scale, can thus be extracted by the naked eye. Comparing antinodal zone boundary spectra for the compressive (LSAO) and tensile (STO) strained systems [Fig. 1e] reveals a softening of about 60 meV in the STO system. To first order, the magnetic-exchange interaction $2J_{eff}$ is setting the antinodal ZB magnon energy scale[11,12]. Without any sophisticated analysis, we thus can conclude that the magnetic-exchange interaction of LCO thin films can be tuned by strain. At the nodal ZB this effect is much

**Table 1 Lattice and model parameters for the different $La_2CuO_4$ film systems**

| Sample | $h$ [nm] | $a$ [Å] | $c$ [Å] | $t$ [meV] exp. | $-t'/t$ exp. | $t$ [meV] DFT | $-t'/t$ DFT | $-t''/t'$ DFT | $U$ [eV] cRPA | $v$ [eV] cRPA |
|---|---|---|---|---|---|---|---|---|---|---|
| LCO/STO | 7–8 | 3.905 | 12.891 | 460.5 | 0.389 | 369.6 | 0.0908 | −0.044 | 1.92 | 12.76 |
| LCO/LSAT | 7–8 | 3.868 | 12.981 | 488.9 | 0.387 | 395.0 | 0.0907 | 0.165 | 2.05 | 13.06 |
| LCO/NGO | 17–19 | 3.864 | 13.077 | 483.6 | 0.388 | 416.1 | 0.0910 | 0.335 | 2.12 | 13.24 |
| LCO/LSAO | 18–19 | 3.756 | 13.195 | 613.2 | 0.422 | 473.7 | 0.0917 | 0.640 | 2.60 | 14.25 |
| Bulk LCO | — | 3.803 | 13.156 | — | — | 443.7 | 0.0915 | 0.510 | 2.40 | 13.86 |
| "Artificial LCO film" | — | 3.842 | 13.105 | — | — | 417.9 | 0.0917 | 0.361 | 2.25 | 13.54 |

Note: Thickness $h$ of the thin films (measured by x-ray diffraction) for substrates as indicated. For the films and bulk LCO, $a$ indicates the room temperature substrate and average in-plane lattice parameter, respectively. The $c$-axis lattice parameters were measured directly by x-ray diffraction (room temperature) on our films whereas for bulk LCO, the literature value is given[44]. For the "Artificial LCO film", $c$-axis was interpolated from the measured samples assuming an in-plane lattice parameter of 3.842 Å. Values of $t$ and $t'$ obtained from the fit using a Hubbard model with $U/t = 9$, $Z = 1.219$ (quantum renormalisation factor)[10,19], and $t'' = -t'/2$. The corresponding theoretical DFT and cRPA results are also included. DFT hopping parameters were obtained using an effective single-band model. Both the screened ($U$) and bare ($v$) interaction increase with in-plane strain within the cRPA methodology. The substrate lattice parameters are taken from refs. [34,50]. Source data are provided as a Source Data file

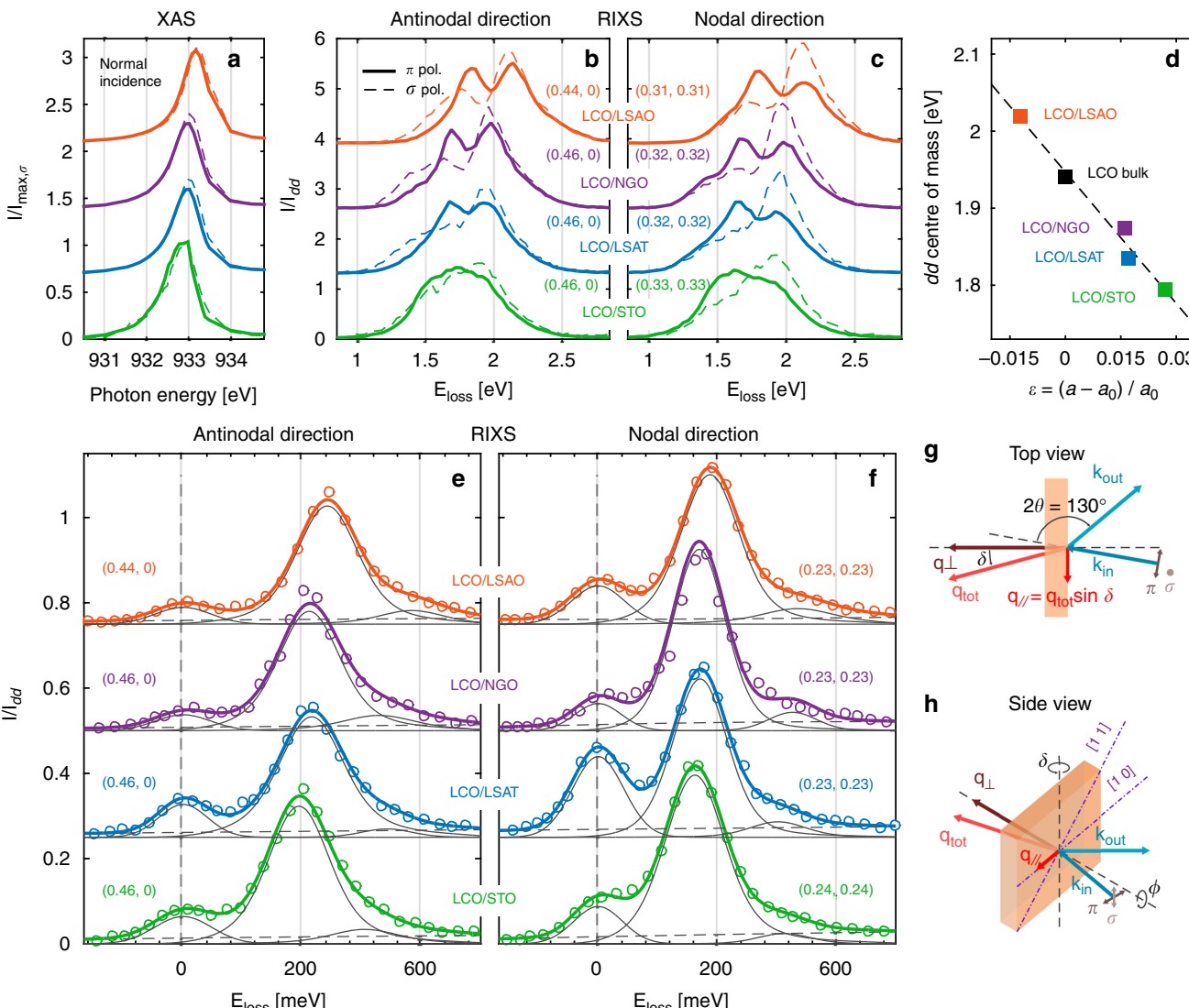

**Fig. 1** Strain-dependent XAS and RIXS spectra of La$_2$CuO$_4$ films. **a** Normal incidence ($\delta = 25°$) XAS spectra recorded around the copper $L_3$-edge on La$_2$CuO$_4$ films on different substrates as indicated. **b–f** display grazing incident copper $L_3$-edge RIXS spectra. In **b**, **c**, the $dd$ excitations are shown for different momenta as indicated. For **a–c**, solid (dashed) lines indicate use of $\pi$ ($\sigma$) polarised incident light. **d** displays the "centre of mass" of the $dd$ excitations vs. strain $\varepsilon$, for samples as indicated. Each (thin film) point is an average "centre of mass" value of all the spectra in **b**, **c**. The bulk La$_2$CuO$_4$ value is extracted from ref. [8]. **e**, **f** present the low-energy part of RIXS spectra (circular points) with four-component (grey lines) line-shape fits (see text). Notice that the different film systems have, naturally, different elastic components. For visibility all curves in **a–e** have been given an arbitrary vertical shift. **g**, **h** illustrate schematically the scattering geometry. Source data are provided as a Source Data file

less pronounced [see Fig. 1f], suggesting a strain dependent zone-boundary dispersion. Therefore, the central experimental observations, reported here, are that the crystal-field environment, the magnetic exchange interaction and the magnon zone-boundary dispersions are tunable through strain.

**Magnon dispersion**. To extract the magnetic-exchange interaction $2J_{\rm eff}$ in a more quantitative fashion, three steps are taken. First, a dense grid of RIXS spectra has been measured along the nodal and antinodal directions in addition to a constant-$|q_{//}|$ trajectory connecting the two [see inset of Fig. 2c]. Second, fitting these spectra allows extracting the full magnon dispersion for all the film systems. Finally, these dispersions are parametrised using strong-coupling perturbation theory for the Hubbard model to extract the effective magnetic exchange couplings.

Compilations of nodal and antinodal RIXS spectra are shown in Fig. 2a, b and Supplementary Fig. 1 The magnon excitations remain clearly visible even near the zone centre where elastic scattering is typically enhanced. To extract the magnon dispersion, such spectra were fitted using a Gaussian line shape for the elastic scattering and a quadratic function for the weak background. The width of the elastic Gaussian is a free parameter in order to account for a small phonon contribution. Two antisymmetric Lorentzian functions[13–15] for the magnon and the small multimagnon signals were adopted and convoluted with the experimental resolution function. The quality of the fits can be appreciated from Figs. 1e, f and 2a, b. Generally, the magnon width was found to be comparable to the experimental resolution and independent of momentum, suggesting that the line shape is resolution limited. In contrast to doped systems[16,10], magnons of the LCO thin films have a negligible damping and hence the

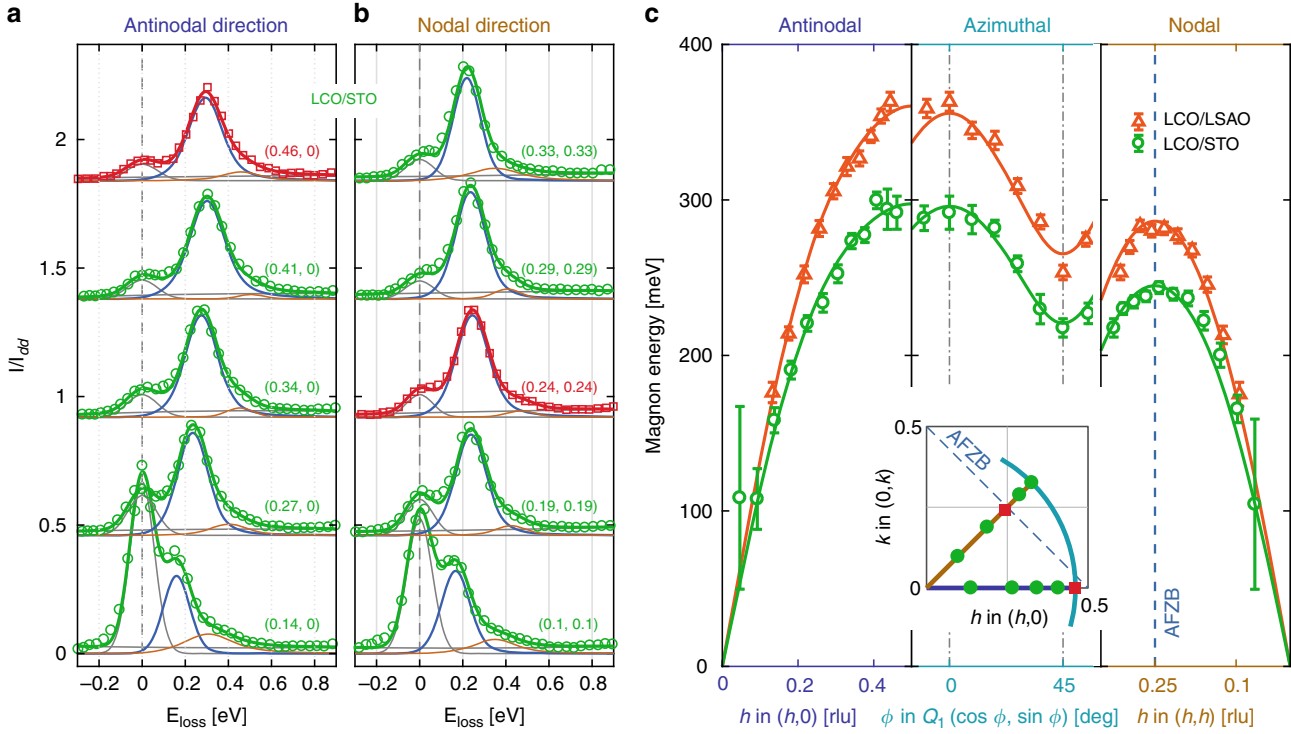

**Fig. 2** Magnon dispersions of La$_2$CuO$_4$ thin films. **a**, **b** display raw RIXS spectra recorded on the LCO/STO thin film system, along the antinodal [1 0] and nodal [1 1] directions, respectively. Red curves represent the data close to the antiferromagnetic zone boundary (AFZB) as shown in the inset in **c**. Solid lines are fits to the data (see text for detailed description). Notice that elastic scattering is, as expected, enhanced as the specular condition (0,0) is approached. In (**c**) magnon dispersions of LCO/LSAO and LCO/STO, extracted from fits to raw spectra as in **a**, **b**, along three different momentum trajectories (see solid lines in the inset) are presented. Solid lines through the data points are obtained from two-dimensional fits using Hubbard model (see the main text). Error bars are three times the standard deviations extracted from the fits. In (**c**) $Q_1$ takes different values for each compound due to slightly different incident energies and in-plane lattice parameters, resulting in 0.4437 (0.4611) for LCO/LSAO (LCO/STO). Source data are provided as a Source Data file

pole of the fitted excitation coincides essentially with the peak maxima.

The extracted magnon dispersions of LCO/STO and LCO/LSAO are displayed in Fig. 2c. The analysis confirms that the magnon bandwidths are significantly different for the two systems. Near the (1/2, 0) zone-boundary point, the LCO/LSAO magnon reaches about 360 meV whereas for LCO/STO a comparative softening of 60 meV is found [Fig. 2c]. This softening is less pronounced near the (1/4, 1/4) zone-boundary point [Fig. 2c], demonstrating that the zone boundary dispersion is also strain dependent [see right inset of Fig. 3c]. Our results thus show a larger ZB dispersion for the LCO/LSAO system.

## Discussion
Different theoretical models have been applied to analyse RIXS spectra of the cuprates. Many of these approaches are purely numerical starting either from a metallic or localised picture[17,18]. To parameterise experimental results, analytical models are useful. The Hubbard model has, therefore, been frequently used to describe the magnon dispersion of La$_2$CuO$_4$[10,11,12,19,20]. We employ a $U - t - t' - t''$ single-band Hubbard model, since $t'$ and $t''$ hopping integrals have previously been shown relevant to account in detail for the magnetic dispersion[19,20]. By mapping onto a Heisenberg Hamiltonian, an analytical expression (see Methods section) for the magnon dispersion $\omega(q)$[10,19,20], has been derived.

Before fitting our results, it is useful to consider the ratios $U/t$, $t'/t$ and $t''/t'$ for single-layer cuprate systems. In-plane strain will

enhance oxygen-$p$ to copper-$d$ orbital hybridisations and hence the effective nearest-neighbour hopping $t$ in the one-band Hubbard model description[21]. This trend can be calculated from approximate numerical methods such as density functional theory (DFT) [see Fig. 3b]. Besides this increase in band width, oxygen-$p$ orbitals are concomitantly pushed down [Supplementary Fig. 4] and the $e_g$ splitting is expected to increase. Indeed, as shown in Fig. 1d, the RIXS $dd$ excitations ("centre of mass") are pushed to higher energies upon compressive strain, which is consistent with an enhanced $e_g$ splitting. This tendency of states moving away from the Fermi-level is generally expected to diminish their ability to screen the Coulomb interaction. Its local component—the Hubbard $U$—quantifies the energetic penalty of adding a second electron to the half-filled effective $d_{x^2-y^2}$ orbital. To a good approximation the evolution of this process is accessible by tracking the Cu $2p^63d^9 \rightarrow 2p^53d^{10}$ XAS resonance. As seen in Fig. 1a and 3a, we find the copper $L$-edge resonance to shift notably upwards under in-plane compression. This strongly suggests that the energy cost for double occupancies—and hence the Hubbard $U$—increases under compressive strain, confirming the above rationale.

Beyond the suggested impact on screening, pressure or strain also modify the localisation of the effective $d_{x^2-y^2}$ orbital. As a basis-dependent quantity, the Hubbard $U$ is sensitive also to this second mechanism[22]. To corroborate our experimental finding for the effective Coulomb interaction under in-plane strain, we therefore carried out cRPA calculations for La$_2$CuO$_4$ that include both screening and basis-localisation effects[22–25], (see Methods section). We stress that cRPA is an approximate numerical

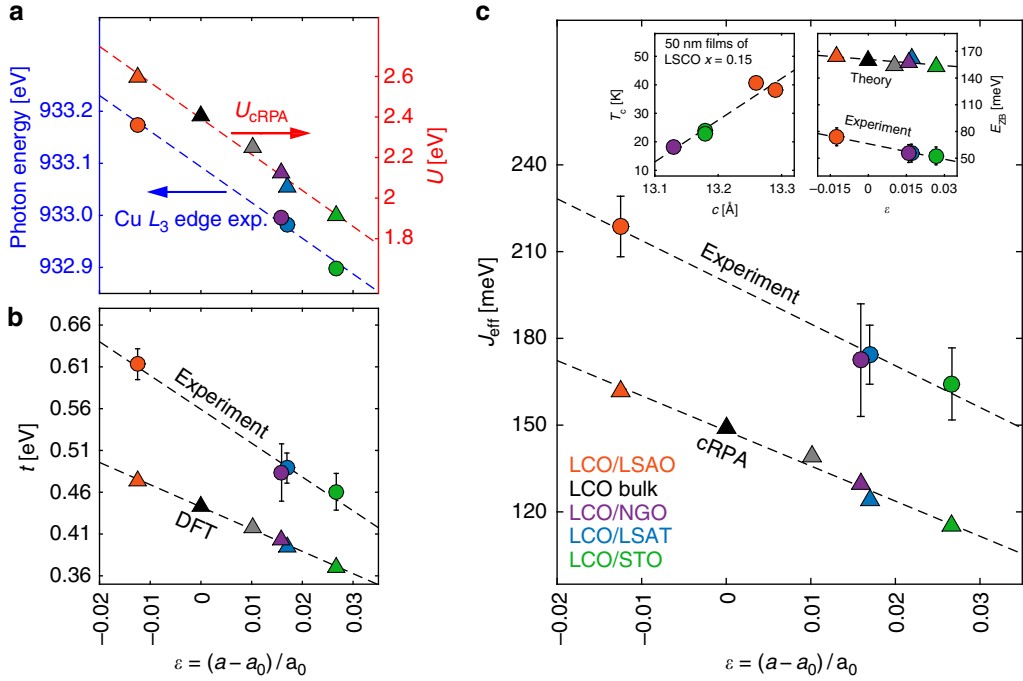

**Fig. 3** Cuprate energy scales vs. strain. In (**a**) XAS at Cu $L_3$-edge resonances (left) and the theoretical results for Coulomb interaction $U$ (right) are presented. Experimental and theoretical derived hopping parameters $t$ are presented in **b**, as indicated. Notice that $t$ is not scaling with the copper-oxygen bond length $r^{-\alpha}$ with $\alpha = 6-7$ as sometimes assumed[51]. $J_{eff}$ as a function of $\varepsilon = (a - a_0)/a_0$ (where $a_0$ is the in-plane bulk lattice parameter) is presented in **c** for both theoretical and experimental results. Zone-boundary dispersion $E_{ZB}$, extracted from the Hubbard model, for experimental and theoretical parameters, are presented in the right inset of **c** as a function of strain $\varepsilon$. Superconducting transition temperature $T_c$ as a function of out-of-plane lattice constant $c$—for optimally doped LSCO thin films (see Supplementary Table 1)—is presented in the left inset in **c**. Colour code for the data points in the figure refers to the one shown in **c**. The error bars for the experimental data are standard deviations extracted from the fits. The theoretical value corresponding to $\varepsilon \approx 0.01$ (gray symbols) is an artificial sample as described in Table 1. Source data are provided as a Source Data file

approach. It is known that correlation effects are underestimated when using only the static limit of the Hubbard interaction[26]. Indeed the cRPA obtained $U \approx 5t$ [see Fig. 3a, b and Table 1] is below the expected bandwidth-controlled threshold value. We therefore focus on the relative trends produced by the cRPA. As shown in Fig. 3a, the simulation indeed predicts the Hubbard $U$ to increase with compressive strain. This confirms the above rationale and thus enables us to interpret the XAS Cu $L_3$-edge as a proxy for the variation of the screened Coulomb interaction $U$. The fact that both $U$ and $t$ increase linearly with compressive strain [Fig. 3a, b] leads us to the Ansatz that the ratio $U/t$ is approximately constant. In the following analysis of the experimental data, we therefore assume $U/t = 9$[19] (and $t''/t' = -0.5$[27]).

In this fashion, our Hubbard model effectively depends only on $t$ and $t'$, that constitute our fitting parameters. On a square lattice, one would expect $t'/t$ to remain approximately constant as a function of strain. Indeed, fitting the magnon dispersions, yields that $t$ increases with reduced lattice parameter [Fig. 3b] while $t'/t \approx -0.4$ (see Table 1). This value of $t'/t$ is reasonably consistent with ARPES and LDA derived band structures of the most tetragonal single-layer cuprate systems Hg1201 and Tl2201[28–31]. Single-band tight-binding models, applied to LSCO, have found significantly lower values of $t'/t$[27,32,33]. However, when including hybridisation between the $d_{x^2-y^2}$ and $d_{z^2}$ orbitals, existing in LSCO, one again finds $t'/t \approx -0.4$[31,28,29]. The described variation of the hopping $t$ and the Hubbard $U$ translates into a pressure-dependent magnetic exchange interaction $J_{eff}$ when mapping the Hubbard model (at strong coupling) into a Heisenberg Hamiltonian: $J_{eff} = 4t^2/U - 64t^4/U^3$. [see Eq. (6)]. Since $U/t \sim$ const., it is therefore expected that $J_{eff}$ scales with $t$.

Indeed, as directly visible from the magnon dispersion and our cRPA calculations [Fig. 3c], $J_{eff}$ increases linearly when going from tensile ($\varepsilon > 0$) to compressive strain ($\varepsilon < 0$).

In LSCO system superconductivity emerges upon hole doping. It is known that for LSCO, the highest $T_c$ is reached when thin films are grown on LSAO substrates[34,35]. Although higher $T_c$ has been linked to larger $c$-axis parameter [see left inset of Fig. 3c], the physical origin of this effect has remained elusive. In-plane strain also tunes the $c$-axis lattice parameter through the Poisson ratio[36]. The observed evolution of the $dd$ excitations [Fig. 1d] is consistent with a compressive strain-induced enhancement of the $e_g$ splitting. It has been argued that this orbital distillation (avoidance of $d_{x^2-y^2}$ and $d_{z^2}$ hybridisation) is beneficial for superconductivity[28,29,31]. The $e_g$ splitting might also indirectly increase $T_c$ by changing the screening of the local Coulomb interaction $U$, as described above. Antiferromagnetic interactions are a known source for $d$-wave Cooper pairing[37]. A link between $J_{eff}$ and $T_c$ is therefore expected in the large $U/t$ limit[38–40]. Here, we have explicitly demonstrated how the important energy scale $J_{eff}$ can be tuned through strain. This direct connection between lattice parameters and the magnetic exchange interaction in Mott insulating La$_2$CuO$_4$ provides an engineering principle for the optimisation of high-temperature cuprate superconductivity.

Our study highlights the power of combining oxide molecular beam epitaxial material design with synchrotron spectroscopy. In this particular case of La$_2$CuO$_4$ thin films, it is shown how Coulomb and antiferromagnetic exchange interactions can be artificially engineered by varying the film substrate. In this

fashion, direct design control on the Mott insulating energy scales, constituting the starting point for high-temperature superconductivity, has been reached. It would be of great interest to apply this strain-control rationale to doped single-layer $HgBa_2CuO_{4+x}$ and $Tl_2Ba_2CuO_{6+x}$ cuprate superconductors to further enhance the transition temperature $T_c$.

## Methods

**Film systems.** High quality $La_2CuO_4$ (LCO) thin films were grown using Molecular Beam Epitaxy (MBE), on four different substrates: $(001)_c - SrTiO_3$ (STO), $(001)_c - (LaAlO_3)_{0.3}(Sr_2TaAlO_6)_{0.7}$ (LSAT), $(001)_{pc} - NdGaO_3$ (NGO) and $(001)_c - LaSrAlO_4$ (LSAO). Comparable LCO film thickness, for STO and LSAT and for NGO & LSAO, were used (Table 1). For such thin films, the in-plane lattice parameter $a_{film}$ is set by the substrate lattice $a$ indicated in Table 1. Compared to bulk LCO, the substrate STO, LSAT and NGO induce tensile strain whereas LSAO generates compressive strain. Film thicknesses were extracted from fit to the $2\theta$ scans (see Supplementary Fig. 2 using an x-ray diffraction tool as in ref. [41].

**Spectroscopy experiments.** X-ray absorption spectroscopy (XAS) and resonant inelastic x-ray scattering (RIXS) were carried out at the ADRESS beamline[42,43], of the Swiss Light Source (SLS) synchrotron at the Paul Scherrer Institut. All data were collected at base temperature (~20 K) of the manipulator under ultra high vacuum (UHV) conditions, $10^{-9}$ mbar or better. RIXS spectra were acquired in grazing exit geometry with both linear horizontal ($\pi$) and linear vertical ($\sigma$) incident light polarisation with a scattering angle $2\theta = 130°$ [see Fig. 1g,h]. An energy resolution half width at half maximum (HWHM) of 68 meV—at the Cu $L_3$ edge—was extracted from the elastic scattering signal. Momentum $q = q_{//} = (h, k)$ is expressed in reciprocal lattice units (rlu).

**Hubbard model.** A single-band Hubbard model is adopted in the present study. Being important to consider a second-neighbour hopping integral—for $La_2CuO_4$ compound[19,20]—in order to fully describe the magnon dispersion relation[11,12], we consider the following Hamiltonian:

$$H = -t \sum_{\langle i,j \rangle, \sigma} c_{i,\sigma}^\dagger c_{j,\sigma} - t' \sum_{\langle\langle i,j \rangle\rangle, \sigma} c_{i,\sigma}^\dagger c_{j,\sigma}$$
$$- t'' \sum_{\langle\langle\langle i,j \rangle\rangle\rangle, \sigma} c_{i,\sigma}^\dagger c_{j,\sigma} + U \sum_i n_{i,\uparrow} n_{i,\downarrow} \quad (1)$$

where $t$, $t'$ and $t''$ are the first-nearest, second-nearest and third-nearest-neighbour hopping integrals; $U$ is the on-site Coulomb interaction integral; $c_{i,\sigma}^\dagger$ and $c_{i,\sigma}$ are the creation and annihilation operators at the site $i$ and spin $\sigma = \uparrow, \downarrow$; and $n_{i,\sigma} \equiv c_{i,\sigma}^\dagger c_{i,\sigma}$ is the density operator at the site $i$ with spin $\sigma$. The sum (for the hopping process) is done over the first-nearest $\langle \star \rangle$, second-nearest $\langle\langle \star \rangle\rangle$ and third-nearest neighbour sites $\langle\langle\langle \star \rangle\rangle\rangle$.

Using this Hamiltonian at strong coupling it is possible to obtain[19,20], a magnon dispersion of the form:

$$\omega(\mathbf{q}) = Z\sqrt{A(\mathbf{q})^2 - B(\mathbf{q})^2}. \quad (2)$$

The momentum dependence of $A$ and $B$ can be expressed in terms of trigonometric functions

$$P_j(h, k) = \cos jha + \cos jka$$
$$X_j(h, k) = \cos jha \cos jka$$
$$X_{3a}(h, k) = \cos 3ha \cos ka + \cos ha \cos 3ka$$

such that[10]:

$$A = 2J_1 + J_2(P_2 - 8X_1 - 26) + 2J_1'(X_1 - 1)$$
$$+ \left[ J_1'' - \frac{8J_1}{U^2}(-t'^2 + 4t't'' - 2t''^2) \right](P_2 - 2)$$
$$+ 2J_2'(-2P_2 + 4X_1 + X_2 - 1) \quad (3)$$
$$+ \frac{2J_1'J_1''}{U}(5P_2 + 2X_1 - 3X_2 - X_{3a} - 7)$$
$$+ J_2''(4P_2 + P_4 - 8X_2 - 2)$$

and

$$B = -J_1P_1 + 16J_2P_1$$
$$- \frac{4J_1}{U^2}\left[(6t'^2 - t't'')(X_1 - 1) + 3t''^2(P_2 - 2)\right]P_1 \quad (4)$$

where $J_1 = \frac{4t^2}{U}$, $J_2 = \frac{4t^4}{U^3}$, $J_1' = \frac{4t'^2}{U}$, $J_2' = \frac{4t'^4}{U^3}$, $J_1'' = \frac{4t''^2}{U}$ and $J_2'' = \frac{4t''^4}{U^3}$. When neglecting higher order terms (i.e., terms in $J_2'$, $J_2''$ and $J_1'J_1''$) and considering $t'' = -t'/2$, it is

possible to obtain an approximated solution for the zone-boundary dispersion $E_{ZB}$[10]:

$$\frac{E_{ZB}}{12ZJ_2} \approx 1 + \frac{1}{12}\left(112 - \frac{J_1}{J_2}\right)\left(\frac{t'}{t}\right)^2, \quad (5)$$

if:

$$\frac{U}{t} \geq \sqrt{\frac{28 + 112\left(\frac{t'}{t}\right)^2}{2 + 3\left(\frac{t'}{t}\right)^2}}, \quad \text{and} \quad \left|\frac{t'}{t}\right| \lesssim 0.686.$$

Furthermore, it is possible to see[19], that with such a model, which considers also the cyclic hopping terms, the effective exchange interaction can be written as:

$$J_{eff} = 4\frac{t^2}{U} - 64\frac{t^4}{U^3} \quad (6)$$

if considering only the first neighbour hopping $t$.

**DFT and cRPA calculations.** We compute the electronic structure of tetragonal bulk $La_2CuO_4$ for lattice constants and atomic positions corresponding to the experimentally investigated thin films (see Table 1). For simplicity, tetragonal structures were considered with the ratio between copper to apical oxygen $d_{O2}$ (copper to lanthanum $d_{La}$) distance and the $c$ axis kept constant to the bulk values $d_{O2}/c = 0.18(4)$ ($d_{La}/c = 0.36(1)$)[44]. We use a full-potential linear muffin-tin orbitals (FPLMTO) implementation[45] in the local density approximation (LDA) and construct maximally localised Wannier functions[46] for the Cu $d_{x^2-y^2}$ orbital. Hopping elements $t$, $t'$ and $t''$ are then extracted by fitting a square-lattice dispersion to high-symmetry points. Next, the static Hubbard $U = U(\omega = 0)$ is computed using the constrained random-phase approximation (cRPA)[47] in the Wannier setup[48] for entangled band-structures[49]. Finally, the effective magnetic exchange interaction $J_{eff}$ is determined using the strong-coupling expression Eq. (6).

The above procedure is an approximate way to account for the screening of the Coulomb interaction $\nu$ provided by the electronic degrees of freedom that are omitted when going to a description in terms of an effective one-band Hubbard (and, ultimately, Heisenberg) model. In other words, $\nu$ has to be screened by all particle-hole polarisations that are not fully contained in the subspace spanned by the $d_{x^2-y^2}$ Wannier functions that define the low-energy model. In cRPA, this partial polarisation is computed within RPA, meaning that bare particle-hole bubble diagrams (Lindhard function) are summed up to all orders in the interaction. Constraining the polarisation comes with the benefit that it is precisely the left-out low-energy excitations that display the most correlation effects, potentially leading to important vertex corrections beyond the RPA. Indeed, solving the many-body model that we are setting up through the hoppings $t$, $t'$, etc. and the Hubbard $U$ would require approaches beyond the RPA.

Let us briefly describe how pressure can modify the partially screened local Coulomb interaction $U$: First, pressure-induced changes in hoppings and crystal-fields modify the solid's polarisation (dielectric function) and, hence, how efficiently the Coulomb interaction is screened. This effect is very material specific and can lead to both, an enhancement or a diminishing of $U$[23-25]. Second, the parameters of the Hubbard model are basis-dependent quantities. As a result the matrix element $U$ also depends on the extent in real-space of the $d_{x^2-y^2}$-derived Wannier basis. Quite counter-intuitively, a pressure-induced delocalisation of Wannier functions generally leads to increased local interactions[22]. This trend can be illustrated by looking at the pressure evolution of the matrix element of the bare (unscreened) Coulomb interaction $\nu = e^2/r$ in the Wannier basis: Indeed, as reported in Table 1, $\nu$ increases with shrinking lattice constant. In our case of tetragonal $La_2CuO_4$, both effects (screening and basis localisation) promote the same tendency: an increase of the Hubbard $U$ under compression.

## Data availability

All experimental data are available upon request to the corresponding authors. The source data underlying Figs. 1–3, Table 1, Supplementary Figs. 1–4 and Supplementary Table 1 are provided as a Source Data file.

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

## Acknowledgements

O.I., M.H. and J.C. acknowledge support by the Swiss National Science Foundation under grant No. BSSGI0_155873 and through the SINERGIA network Mott Physics Beyond the Heisenberg Model. D.E.N., E.P., Y.T. and T.S. acknowledge support by the Swiss National Science Foundation through its Sinergia network Mott Physics Beyond the Heisenberg Model MPBH (Research Grant CRSII2_160765/1) and the NCCR MARVEL (Research Grant 51NF40_141828). This work was performed at the ADRESS beamline of the SLS at the Paul Scherrer Institut, Villigen PSI, Switzerland. We thank the ADRESS beamline staff for technical support. C.A. and M.R.B. were is supported by Air Force Office of Scientific Research grant No. FA9550-09-1-0583. N.E.S. and H.M.R. acknowledge the Swiss National Science foundation under grant No. 200021-169061. W.W. and N.B.C. were supported by the Danish Center for Synchrotron and Neutron Science (Dan-Scatt). K.M.S. and H.I.W. were supported by the Air Force Office of Scientific Research grant No. FA9550-15-1-0474.

## Author contributions

H.I.W., C.A., C.L., M.G., M.R.B. and K.M.S. grew and characterised the La$_2$CuO$_4$ thin films. O.I., M.H., W.W., N.B.C., D.E.N., E.P., Y.T., T.S. and J.C. executed the XAS and RIXS experiments. O.I., W.W. and N.B.C. performed the RIXS and XAS data analysis. N.E.S. and H.M.R. developed the Hubbard model. J.M.T. carried out the DFT and cRPA calculations. All authors contributed to the manuscript.

## Additional information

**Competing interests:** The authors declare no competing interests.

