## [Peer Review File · Nature Communications]

Reviewers' comments:

Reviewer #1 (Remarks to the Author):

The paper by O. Ivashko et al. discusses the dependence of the basic parameters of the cuprate models on the compressive strain. To this end the Authors perform a detailed analysis of the RIXS and XAS results on La₂CuO₄ grown on substrates with different lattice parameters. In my opinion the two main results are as follows: (i) the Hubbard U increases with the compressive strain, and (ii) the spin exchange J increases with the compressive strain. These results, in particular (i), are very interesting and in principle deserve publication in Nature Communications. Unfortunately, I am not sure about the conclusions concerning the dependence of the Hubbard U on pressure:

Firstly, I would appreciate if the Authors can explain what is the physical origin of the increasing Hubbard U upon pressure. How are the cRPA calculations performed? To what extent we can trust such approximate calculations applied to an (in principle) unsolvable physics problem?

Secondly, if I understand it correctly, this result is experimentally obtained from the dependence of the energy of the Cu L3 edge XAS resonance on the strain. Let me, however, try to offer an alternative explanation to the observed behaviour:

(a) The increased pressure leads to the larger Cu-O hoppings and hence stronger covalency effects.

(b) The latter effect lowers the energy of the single hole in the ground state $3d^9 x^2-y^2$ orbital which strongly mixes with the hybridising nearest neighbour oxygen orbitals.

(c) On the other hand, the $3d^{10}$ configuration is not sensitive to the effect discussed in (a).

(d) Combining (b) and (c) once can conclude that the energy one needs to pay in the XAS resonance [Cu ($2p^6 3d^9$) \rightarrow Cu ($2p^5 3d^{10}$)] should increase.

Could the Authors comment on the above explanation? Couldn't this be more plausible than a simple change in the effective U in the single-band Hubbard model?

Minor suggestion: The form of the discussion presented in the first two paragraphs of page 5 can perhaps be a bit improved, since it is not fully clear to me what were the particular steps taken in the fitting procedures (which parameters were constant and which varied, in order to achieve results presented in Fig. 3).

Reviewer #2 (Remarks to the Author):

In this manuscript, O. Ivashko et al. present an impressive set of XAS/RIXS experimental data collected at the L3 edge of copper in a series of La₂CuO₄ strained thin films. The variation of lattice parameters leads to a significant modulation of the electronic structure which is captured by the evolution of spectral features. The authors fit the data using a $t-t''-U$ Hubbard model (similar to their previous publication PRB 95, 214508, 2017) and support the interpretation with DFT/cRPA calculations. The paper illustrates how the superconducting transition temperature scales up with increasing compressive strain; this result confirms previous observations by Abrecht et al. PRL 91, 057002, 2003 (not cited). The originality of the present manuscript resides in the detailed link established to the NN hopping integrals, Coulomb and effective exchange interactions.

The data is presented in a very clear manner. The manuscript is well written and will eventually provide valuable reading to a broad community.

The authors managed to build up a strong scientific case and therefore I support the publication of the manuscript in Nat. Commun. after addressing the following points:

(1) lines 79-80: the authors claim "we demonstrate that U/t is essentially untouched by strain". The statement is certainly supported by similar trends calculated for U and t . However the U/t ratio was simply kept fixed when fitting the experimental data, as in PRB 95, 214508, 2017. Would a fit with different U/t ratio lead to unphysical results? As the answer to this question is unclear the claim should either be softened or restricted to the computational-based argument.

(2) figure 3: t and J_{eff} obtained from calculations are situated well outside the error bars of values derived from experiment. Same statement is valid when comparing $U=9*t_{\text{(exp)}}$ from Table I with U_{cRPA} values in Fig. 3. This aspect certainly requires some further explanations/analysis.

(3) T. Nomura published in 2015 a theoretical study of La_2CuO_4 using rather similar theoretical tools (Journal of the Physical Society of Japan 84, 094704, 2015, not cited). It might be interesting to learn about the agreement/disagreement with the data in the present manuscript.

(4) As Nat. Commun. addresses to a large public, the authors might strengthen the manuscript with an appropriate outlook.

Two minor suggestions concerning the form:

- the reference list should be revised as some journal title abbreviations are wrong (see [2]), some are not abbreviated at all (see [8]) and so forth.
- the name of chemical elements should not begin with capital letters (line 355).

Reviewer #3 (Remarks to the Author):

The manuscript of Ivashko et al (NCOMMS-18-23431-T) deals with magnetic fluctuations on parent compounds of superconducting copper oxide thin films, namely $\text{La}_{2-x}\text{CuO}_{4-x}$ materials grown on various substrates to increase or decrease the copper-oxygen distance and then orbitals overlap. X-ray absorption (XAS) and resonant inelastic x-ray scattering (RIXS) spectroscopy are used to determine the magnetic interactions. The authors have performed an impressive set of measurements. A clear effect is seen in RIXS, spin-waves dispersions are evolving as expected from the copper-oxygen distance. The authors however do not compute how the magnetic interactions should change with that distance, a typical expectation is $J \propto 1/r^{6-7}$. Is it the case?

I found the manuscript straightforward to read. Overall, this could be an interesting and important paper, since it shows the effectiveness of epitaxial strain for tuning of the electronic excitations. However, I am afraid that the authors offer an opaque physical picture. In essence, the paper is rather conventional. It is not surprising that the magnetic interactions increase when the copper-oxygen orbitals overlap increases. The discussion based on simple single-band Hubbard model and DFT calculations is not very elaborate, the theoretical considerations do not make a significant breakthrough. How should we trust a model which, for instance, does not catch the ratio between second neighbour and first neighbour hopping integrals t'/t ? This ratio is known to be important to get high- T_c superconductivity when the system is hole doped. Why the authors do not start with a model which properly described the electronic structure?

I would prefer that they would describe the data with Heisenberg model and that they plot the relevant physical parameters (first neighbour J , cyclic interaction,...) versus strain and/or distance. Then, one can appreciate which magnetic interactions is changing.

Further, I consider that they oversimplify the physical picture of superconducting cuprates when they argue high- T_c superconductivity is arising from the Mott physics. I do not share their enthusiasm that their result "provides an engineering principle for optimization of high-temperature cuprate superconductivity." (The title of the manuscript). The value of J_{eff} alone is not controlling high- T_c superconductivity. The authors have a prejudice that the cuprate family based on La_2CuO_4 is the archetypal of single layer superconducting cuprates. It is known that hole-doped La_2CuO_4 exhibits lower superconducting properties compared to other hole-doped single layer cuprate (TI-based or Hg-based). Typically, La_2CuO_4 exhibits a strong structural distortion where the Cu-O-Cu bonding is not at 180° , with a large impact on the super-exchange interactions. This is overlooked in the manuscript. Although the manuscript presents an impressive set of experimental data, I therefore cannot recommend publication in Nature Comm.

The authors should also consider the following questions:

1) Why the RIXS elastic line changes significantly from going from lower momentum to large momentum (fig 2ab) ? Is it due to low energy phonon contribution ? No explanation is given. In figs 1ef, the elastic line changes significantly. Why is that ?

2) Fig S2 shows the scaling of all RIXS datasets that the authors used to say that only one parameter is renormalized when changing the substrate. However, I notice a slight difference along the nodal direction indicating that the renormalization is not uniform.

Reviewer #1 (Remarks to the Author):

Referee 1: The paper by O. Ivashko et al. discusses the dependence of the basic parameters of the cuprate models on the compressive strain. To this end the Authors perform a detailed analysis of the RIXS and XAS results on La₂CuO₄ grown on substrates with different lattice parameters. In my opinion the two main results are as follows: (i) the Hubbard U increases with the compressive strain, and (ii) the spin exchange J increases with the compressive strain. These results, in particular (i), are very interesting and in principle deserve publication in Nature Communications. Unfortunately, I am not sure about the conclusions concerning the dependence of the Hubbard U on pressure:

Authors: We thank the referee for the concise summary and positively inclined recommendation. Below, we address the comments, questions and suggestions from the referee.

Referee 1: Firstly, I would appreciate if the Authors can explain what is the physical origin of the increasing Hubbard U upon pressure.

Authors: Generally, the Hubbard U depends on (a) screening and (b) the real-space extent of the $d_{x^2-y^2}$ orbital. Pressure-induced changes to crystal fields and band-widths can modify the screening potential resulting in either an increase or decrease of U dependent on the exact material. In our case of La₂CuO₄ both (a) and (b) conspire to an increasing U with compressive strain. We appreciate that the referee would like to have simple general (material independent) arguments for a pressure induced increase of U . Unfortunately, the pressure dependence of U is more complicated and must be considered case by case. We have added an addition paragraph to the method section describing on general grounds the pressure effects on U .

Referee 1: Secondly, if I understand it correctly, this result is experimentally obtained from the dependence of the energy of the Cu L3 edge XAS resonance on the strain. Let me, however, try to offer an alternative explanation to the observed behaviour:

(a) The increased pressure leads to the larger Cu-O hoppings and hence stronger covalency effects.

(b) The latter effect lowers the energy of the single hole in the ground state $3d^9 x^2-y^2$ orbital which strongly mixes with the hybridising nearest neighbour oxygen orbitals.

(c) On the other hand, the $3d^{10}$ configuration is not sensitive to the effect discussed in (a).

(d) Combining (b) and (c) once can conclude that the energy one needs to pay in the XAS resonance [Cu ($2p^6 3d^9$) \rightarrow Cu ($2p^5 3d^{10}$)] should increase.

Could the Authors comment on the above explanation? Couldn't this be more plausible than a simple change in the effective U in the single-band Hubbard model?

Authors: The referee provides an explanation of the strain dependent Cu L3 edge XAS resonance without considering Coulomb interaction. In this atomic picture, the energy levels are center-of-mass projections of the band structure. As correctly pointed-out in (a), compressive strain increases the Cu-O hoppings and hence the effective t (bandwidth). However, this increase of kinetic energy is not modifying the energetic difference between one and two electrons in the x^2-y^2 level. This latter cost is related to the potential energy and hence the Coulomb interaction. We are thus arguing that the effects described by point (b)-(d) are implicitly linked to the Coulomb interaction.

In fact, the scenario proposed by the referee provides a simple explanation for U increasing with strain. The Coulomb interaction is defined by the cost of double occupancy in the $d_{x^2-y^2}$ orbital. In our case, it is

thus the difference between $3d^9$ and $3d^{10}$. Considering the single hole ($3d^9$) state, its atomic energy level will decrease with in-plane strain (Referee's point b). As a result, the $3d^9 \rightarrow 3d^{10}$ transition (Referee's point d) and hence U , should as a function of compressive in-plane strain increase in energy. We have included this simple line of arguments to our manuscript. In fact, we have used the referee's suggestion to restructure the discussion sections. All manuscript changes are highlighted with blue-colored text.

In addition to the referee's elegant arguments, we also hint to the similar oxygen K -edge XAS process. As $d_{x^2-y^2}$ and oxygen p -orbitals are hybridizing, Cu-L and O-K edge XAS is in both cases promoting a core electron to the unoccupied density of states. In case of OK XAS, the pre-edge of La_2CuO_4 has been interpreted in terms of the upper Hubbard band. We use the same interpretation for the L-edge.

Referee 1: How are the cRPA calculations performed? To what extent we can trust such approximate calculations applied to an (in principle) unsolvable physics problem?

Authors: To address these questions, we have

- (a) Made a clear title in the method section to the paragraph describing the cRPA calculations.
- (b) Extent the description of the cRPA methodology.
- (c) Stressed directly in the text that the cRPA is an approximative methodology and hence conclusions are only based on trends.

Referee 1: Minor suggestion: The form of the discussion presented in the first two paragraphs of page 5 can perhaps be a bit improved, since it is not fully clear to me what were the particular steps taken in the fitting procedures (which parameters were constant and which varied, in order to achieve results presented in Fig. 3).

Authors: Thanks for the excellent suggestion, we have improved the initial part of the "Discussion" section accordingly. We stress explicitly that t and t' are our two fitting parameters.

Reviewer #2 (Remarks to the Author):

Referee 2: In this manuscript, O. Ivashko et al. present an impressive set of XAS/RIXS experimental data collected at the L3 edge of copper in a series of La_2CuO_4 strained thin films. The variation of lattice parameters leads to a significant modulation of the electronic structure which is captured by the evolution of spectral features. The authors fit the data using a t - t' - U Hubbard model (similar to their previous publication PRB 95, 214508, 2017) and support the interpretation with DFT/cRPA calculations. The paper illustrates how the superconducting transition temperature scales up with increasing compressive strain; this result confirms previous observations by Abrecht et al. PRL 91, 057002, 2003 (not cited). The originality of the present manuscript resides in the detailed link established to the NN hopping integrals, Coulomb and effective exchange interactions.

Authors: The work of Abrecht et al. PRL 91, 057002, 2003 is cited in the revised version of the manuscript.

The data is presented in a very clear manner. The manuscript is well written and will eventually provide valuable reading to a broad community.

The authors managed to build up a strong scientific case and therefore I support the publication of the manuscript in Nat. Commun. after addressing the following points:

Authors: We thank referee 2 for his/her recommendation to publish our work in Nature Communications and for the suggestions to further improve the manuscript.

Referee 2: (1) lines 79-80: the authors claim "we demonstrate that U/t is essentially untouched by strain". The statement is certainly supported by similar trends calculated for U and t . However the U/t ratio was simply kept fixed when fitting the experimental data, as in PRB 95, 214508, 2017. Would a fit with

different U/t ratio lead to unphysical results? As the answer to this question is unclear the claim should either be softened or restricted to the computational-based argument.

Authors: These are good questions and suggestions. We have softened the statement "...essentially untouched...", so that it now reads "... demonstrate that U/t remains approximately constant with in-plane strain."

Referee 2: (2) figure 3: t and J_{eff} obtained from calculations are situated well outside the error bars of values derived from experiment. Same statement is valid when comparing $U=9*t_{\text{(exp)}}$ from Table I with U_{cRPA} values in Fig. 3. This aspect certainly requires some further explanations/analysis.

Authors: This is a good point that is somewhat related to comments from referee 1 and 3. In our revised manuscript, we have stressed much stronger that cRPA is an approximative approach. As a consequence, we are concluding trends rather than absolute values of U and J_{eff} . This fact has been stressed directly in the revised discussion of the main text.

Referee 2: (3) T. Nomura published in 2015 a theoretical study of La_2CuO_4 using rather similar theoretical tools (Journal of the Physical Society of Japan 84, 094704, 2015, not cited). It might be interesting to learn about the agreement/disagreement with the data in the present manuscript.

Authors: We thank the referee for pointing to this reference that is now cited. As this is numerical work, it is not straightforward to compare with our results. However, stimulated by this suggestion and comments from referee 3, we have included a paragraph mentioning the different theoretical approaches to model the RIXS response of La_2CuO_4 . This paragraph also motivates our choice to project the single-band Hubbard model into an analytically solvable Heisenberg Hamiltonian.

Referee 2: (4) As Nat. Commun. addresses to a large public, the authors might strengthen the manuscript with an appropriate outlook.

Authors: This is an excellent suggestion from the referee. We have amended the manuscript with a concluding outlook.

Two minor suggestions concerning the form:

- the reference list should be revised as some journal title abbreviations are wrong (see [2]), some are not abbreviated at all (see [8]) and so forth.
- the name of chemical elements should not begin with capital letters (line 355).

Authors: We have corrected these formatting issues in the revised version of the manuscript.

Reviewer #3 (Remarks to the Author):

Referee 3: The manuscript of Ivashko et al (NCOMMS-18-23431-T) deals with magnetic fluctuations on parent compounds of superconducting copper oxide thin films, namely $\text{La}_{2-x}\text{CuO}_4$ materials grown on various substrates to increase or decrease the copper-oxygen distance and then orbitals overlap. X-ray absorption (XAS) and resonant inelastic x-ray scattering (RIXS) spectroscopy are used to determine the magnetic interactions. The authors have performed an impressive set of measurements. A clear effect is seen in RIXS, spin-waves dispersions are evolving as expected from the copper-oxygen distance. The authors however do not compute how the magnetic interactions should change with that distance, a typical expectation is $J \propto 1/r^6$. Is it the case?

Authors: We assume that r refers to the copper-oxygen distance. The copper-oxygen hopping integral t_{dp} scales as $r^{-\alpha}$ with assumed $\alpha=3-3.5$ (PRB 89, 085113 (2014)). Now, the nearest neighbor hopping t

used in our manuscript is $t \propto t_{dp}^2$. The magnetic exchange interaction J is, to first order, given by t^2/U . In our manuscript, we used the Ansatz $U \propto t$, and hence $J \propto t \propto r^{-(6-7)}$. This is the origin of the expectation mentioned by the referee. Both our experimental data and the DFT calculation suggest a smaller value of α .

I found the manuscript straightforward to read. Overall, this could be an interesting and important paper, since it shows the effectiveness of epitaxial strain for tuning of the electronic excitations.

Authors: We thank the referee 3 for his/her appreciation of our manuscript.

Referee 3: However, I am afraid that the authors offer an opaque physical picture. In essence, the paper is rather conventional. It is not surprising that the magnetic interactions increase when the copper-oxygen orbitals overlap increases. The discussion based on simple single-band Hubbard model and DFT calculations is not very elaborate, the theoretical considerations do not make a significant breakthrough.

Authors: We would like to stress that our work is experimental. Our manuscript is therefore not communicating a theoretical breakthrough. Modeling is used for different purposes. (a) By projecting the single-band Hubbard model into a Heisenberg model, an analytic expression for the magnon dispersion is used to parametrize experimental observables. (b) We have employed DFT and cRPA as approximate numerical models to corroborate the global trends on t , U and J_{eff} as a function of strain as extracted from experiment.

Referee 3: How should we trust a model which, for instance, does not catch the ratio between second neighbour and first neighbour hopping integrals t'/t ?

Authors: In our manuscript, we are not discussing DFT predictions of t'/t . It is, however, done in our Ref. 24, 25 (Sakakibara PRL 2010 and PRB 2012). In general, good agreement between experimental observations and DFT predictions of t'/t is found. Once the role of the d_{z^2} states is included, agreements are also found for La-based cuprates. The universal value of $t'/t = 0.3-0.4$ is also found in the parametrization of our data.

Referee 3: This ratio is known to be important to get high- T_c superconductivity when the system is hole doped.

Authors: Probably the referee is referring to Pavarini *et al.*, PRL 87, 047003 (2001). Here, it is indeed suggested that t'/t is an important parameter for T_c . The more recent works of Sakakibara PRL 2010 and PRB 2012 have, however, demonstrated that once the d_{z^2} orbital is taken into consideration, $t'/t = 0.3-0.4$ for all the hole doped cuprates including Tl2201 and Hg2201. The statement of the referee thus conflicts with the work Sakakibara PRL 2010 and PRB 2012.

Referee 3: Why the authors do not start with a model which properly described the electronic structure?

Authors: To our knowledge there exists no one theory or numerical approach that would give the absolute correct electronic structure of the cuprates. As mentioned above, we use DFT and cRPA to outline trends of t and J_{eff} .

Referee 3: I would prefer that they would describe the data with Heisenberg model and that they plot the relevant physical parameters (first neighbour J , cyclic interaction,...) versus strain and/or distance. Then, one can appreciate which magnetic interactions is changing.

Authors: Although not stated explicitly, our Hubbard modeling is actually projected into the Heisenberg model. In the method section, we therefore did express the magnon dispersion in terms of the magnetic exchange interactions. We are thus using a Heisenberg model where the magnetic exchange interactions

by construction is consistent with the $U-t-t'-t''$ Hubbard model. In the revised version of the manuscript, we have stressed this point stronger by mentioning it both in the main text and the method section. In Fig. 3, $J_{\text{eff}}=4t^2/U-64t^4/U^3$ is plotted versus strain. Since t' and t'' scale with t , higher order interactions will have similar trends.

Referee 3: Further, I consider that they oversimplify the physical picture of superconducting cuprates when they argue high- T_c superconductivity is arising from the Mott physics. I do not share their enthusiasm that their result "provides an engineering principle for optimization of high-temperature cuprate superconductivity." (The title of the manuscript). The value of J_{eff} alone is not controlling high- T_c superconductivity.

Authors: From the referee comments, it seems that he/she implicitly agrees that J_{eff} is one of the parameters influencing T_c . We never intended to claim that it is the only parameter and we don't think that this is expressed by the title. In the light of referee 3's next comment, we have revised the title so that it reads "*Parental Control of Mott-Insulating La_2CuO_4* ". In this fashion, we restrict the title to the material studied and without reference to superconductivity.

Referee 3: The authors have a prejudice that the cuprate family based on La_2CuO_4 is the archetypal of single layer superconducting cuprates. It is known that hole-doped La_2CuO_4 exhibits lower superconducting properties compared to other hole-doped single layer cuprate (TI-based or Hg-based). Typically, La_2CuO_4 exhibits a strong structural distortion where the Cu-O-Cu bonding is not at 180° , with a large impact on the super-exchange interactions. This is overlooked in the manuscript. Although the manuscript presents an impressive set of experimental data, I therefore cannot recommend publication in Nature Comm.

Authors: The referee concludes by pointing out that bulk La_2CuO_4 has orthorhombic distortions in contrast to the tetragonal structured TI- or Hg-based compounds. To this end, it should be mentioned that the TI- and Hg-based systems can't be synthesized with half-filling. More importantly, we are not studying the orthorhombic bulk phase. We investigate thin films strained by their respective substrates. STO, LSAO and LSAT are not imposing orthorhombic strain because $a=b$ in-plane for these substrates. As our films are thin and not significantly relaxed, the lattice should be set by the substrate.

To alleviate the concern of referee 3, we have revised our manuscript in the following ways.

- (a) We have removed the term "prototypical" from the abstract.
- (b) In the outlook proposed by referee 2, we have mentioned the possibility to study doped TI- or Hg-based compounds

Referee 3: The authors should also consider the following questions:

1) Why the RIXS elastic line changes significantly from going from lower momentum to large momentum (fig 2ab)? Is it due to low energy phonon contribution? No explanation is given.

Authors: This effect on the elastic line is observed in virtually all RIXS experiments (bulk and films). In fact, the specular condition (zero momentum) is found by a pronounced maximum in elastic scattering. We have mentioned this fact in the revised manuscript. On the other hand, in grazing conditions, the elastic line is generally suppressed. The phonon contribution to this effect is usually negligible.

Referee 3: In figs 1e,f, the elastic line changes significantly. Why is that?

Authors: In Fig. 1 (e,f), we are comparing RIXS spectra recorded in films grown on different substrates. The films are therefore not identical and hence one would not expect identical elastic scattering. In fact, elastic scattering can vary from sample to sample, due to, for example, different film quality. We have added a comment to the manuscript explaining this fact.

Referee 3: 2) Fig S2 shows the scaling of all RIXS datasets that the authors used to say that only one

parameter is renormalized when changing the substrate. However, I notice a slight difference along the nodal direction indicating that the renormalization is not uniform.

Authors: The effect pointed out by the referee is not statistically significant.

REVIEWERS' COMMENTS:

Reviewer #1 (Remarks to the Author):

I am very grateful to O. Ivashko et al. for such a careful and detailed response to my comments. I would like to recommend the paper for publication in Nature Communications, provided two (very) minor revisions are made in the final version of the manuscript:

(i) In the beginning of the discussion section (page 5 of the paper) it would be better to write, even more explicitly than in the current version, that the "theoretical analysis presented in the paper is based on the single-band Hubbard model". This way it would be clear to the Reader that also alternative explanations, e.g. based on the charge-transfer (=three-band) model, are in principle possible.

(ii) To facilitate reading, the first paragraph of the discussion section (page 5 of the paper), which is pretty long, should be split into two parts.

Reviewer #2 (Remarks to the Author):

The revised manuscript submitted by O. Ivashko et al. addresses a series of questions set by the three referees.

The authors used the provided input in a constructive manner to amend the manuscript. The aspects listed in my assessment of the original contribution led to adequate changes of the paper.

In conclusion I can reconfirm my support for the publication of the manuscript in Nature Communications.

Reviewer #3 (Remarks to the Author):

The authors basically reply to most of my concerns. In particular, they better link their theoretical Hubbard model to the Heisenberg model, relevant to analyze the RIXS data. The title and the discussion part have been appropriately revised. They took back their unnecessary assertions about high-temperature superconductivity, in particular about the archetypical status of La-based cuprates. The paper deserves publication in Nature Communications.

I am however surprised by some of their answers, for instance when they wrote,

Authors: In our manuscript, we are not discussing DFT predictions of t'/t . It is, however, done in our Ref.

24, 25 (Sakakibara PRL 2010 and PRB 2012). In general, good agreement between experimental observations and DFT predictions of t'/t is found. Once the role of the d_{z^2} states is included, agreements

are also found for La-based cuprates. The universal value of $t'/t = 0.3-0.4$ is also found in the parametrization of our data.

As I said previously, the ratio $|t'/t|$ around 0.3-0.4 is indeed important for cuprates. It is an empirical fact. However, I do not see that in their table I of their theoretical DFT calculations.

I'm glad to see that once one includes the d_{z^2} orbitals, proper ratio $|t'/t|$ around 0.3-0.4 can be found theoretically for all cuprates, but later,

Authors: To our knowledge there exists no one theory or numerical approach that would give the absolute correct electronic structure of the cuprates. As mentioned above, we use DFT and cRPA to outline trends of t and J_{eff} .

So, is it possible to describe the correct electronic structure cuprates or not? Why they do not include d_{z^2} orbital in their DFT calculation? They rather prefer work with incorrect t'/t ratio which does not describe their experimental data.

It looks that the authors prefer avoid to reply to the original question. The same can be said about my first question about the variation of J vs the copper-oxygen distance. They repeat in their reply the arguments that it should be $1/r^{\{6-7\}}$ but they do not care about addressing that in the manuscript. They seem to observe a different tendency with smaller α .

These are not blocking points to publish the paper, however it is unsatisfactory that they do not reply directly.

Reviewer #1 (Remarks to the Author):

I am very grateful to O. Ivashko et al. for such a careful and detailed response to my comments. I would like to recommend the paper for publication in Nature Communications, provided two (very) minor revisions are made in the final version of the manuscript:

(i) In the beginning of the discussion section (page 5 of the paper) it would be better to write, even more explicitly than in the current version, that the “theoretical analysis presented in the paper is based on the single-band Hubbard model”. This way it would be clear to the Reader that also alternative explanations, e.g. based on the charge-transfer (=three-band) model, are in principle possible.

(ii) To facilitate reading, the first paragraph of the discussion section (page 5 of the paper), which is pretty long, should be split into two parts.

Authors: We thank referee 1 for his/her suggestions that have been implemented.

Reviewer #2 (Remarks to the Author):

The revised manuscript submitted by O. Ivashko et al. addresses a series of questions set by the three referees.

The authors used the provided input in a constructive manner to amend the manuscript. The aspects listed in my assessment of the original contribution led to adequate changes of the paper.

In conclusion I can reconfirm my support for the publication of the manuscript in Nature Communications.

Authors: We thank referee 2 for reconfirming his/her recommendation.

Reviewer #3 (Remarks to the Author):

The authors basically reply to most of my concerns. In particular, they better link their theoretical Hubbard model to the Heisenberg model, relevant to analyze the RIXS data. The title and the discussion part have been appropriately revised. They took back their unnecessary assertions about high-temperature superconductivity, in particular about the archetypical status of La-based cuprates. The paper deserves publication in Nature Communications.

Authors: We thank referee 3 for his/her recommendation for publication in Nature Communication. Below, we answer the remaining points.

I am however surprised by some of their answers, for instance when they wrote,

Authors: In our manuscript, we are not discussing DFT predictions of t'/t . It is, however, done in our Ref. 24, 25 (Sakakibara PRL 2010 and PRB 2012). In general, good agreement between experimental observations and DFT predictions of t'/t is found. Once the role of the d_{z^2} states is included, agreements are also found for La-based cuprates. The universal value of $t'/t = 0.3-0.4$ is also found in the parametrization of our data.

As I said previously, the ratio $|t'/t|$ around 0.3-0.4 is indeed important for cuprates. It is an empirical fact. However, I do not see that in their table I of their theoretical DFT calculations.

I'm glad to see that once one includes the d_{z^2} orbitals, proper ratio $|t'/t|$ around 0.3-0.4 can be found theoretically for all cuprates, but later,

Authors: To our knowledge there exists no one theory or numerical approach that would give the

absolute correct electronic structure of the cuprates. As mentioned above, we use DFT and cRPA to outline trends of t and J_{eff} .

So, is it possible to describe the correct electronic structure cuprates or not? Why they do not include d_{z^2} orbital in their DFT calculation? They rather prefer work with incorrect t'/t ratio which does not describe their experimental data.

Authors: The t'/t found in the literature are inferred from tight-binding fitting to either experimental or theoretical (DFT) band structure calculations. DFT calculations would, of course, include the d_{z^2} orbital. This is true for our work and for already published DFT calculations. Now, when fitting this calculated band structure, one has a choice to use a single- or two-orbital tight binding model. In Sakakibara PRL 2010 and PRB 2012, it is shown that for Hg1201 this choice doesn't matter as similar values of t'/t are obtained. For LSCO however, due to the hybridization between $d_{x^2-y^2}$ and d_{z^2} orbitals, correct values of t'/t is obtained only for the two-orbital tight-binding model. Now, if one is only interested in providing a parametrization of data, it can still be useful to use a single-band model. This is what we have done in our case. In the revised caption of Table 1, we are now mentioning this explicitly.

It looks that the authors prefer avoid to reply to the original question. The same can be said about my first question about the variation of J vs the copper-oxygen distance. They repeat in their reply the arguments that it should be $1/r^{\{6-7\}}$ but they do not care about addressing that in the manuscript. They seem to observe a different tendency with smaller α .

Authors: Indeed in our reply, we demonstrated under which assumptions the $1/r^{\{6-7\}}$ occurs. It is based on the assumption, that the copper-oxygen hopping integral t_{dp} scales as $r^{-\alpha}$ with $\alpha=3-3.5$ (PRB 89, 085113 (2014)). We are now mentioning in Figure caption 3 that our results are not consistent with this particular assumption.

These are not blocking points to publish the paper, however it is unsatisfactory that they do not reply directly.